# The Association of IL-17 and PlGF/sENG Ratio in Pre-Eclampsia and Adverse Pregnancy Outcomes

**DOI:** 10.3390/ijerph20010768

**Published:** 2022-12-31

**Authors:** Dorota Darmochwal-Kolarz, Anita Chara

**Affiliations:** 1Medical College, University of Rzeszow, 35-959 Rzeszow, Poland; 2Department of Obstetrics and Gynaecology, 27-600 Sandomierz, Poland

**Keywords:** interleukin-17 (IL-17), endoglin (sENG), FGR (fetal growth restriction), placental growth factor (PlGF), PlGF/sENG ratio, pre-eclampsia, pregnancy

## Abstract

The aim of the study was to assess the role of concentrations of interleukin-17 (IL-17), placental growth factor (PlGF) and soluble endoglin (sENG), as well as the PlGF/sENG ratio in pregnancy complicated by pre-eclampsia (PE) and normal pregnancy. The concentrations of IL-17, PlGF and sENG were measured with the use of immunoenzymatic methods. The concentrations of IL-17 were significantly higher in PE patients when compared to control patients. In the group of patients with PE, the levels of IL-17 positively correlated with systolic blood pressure. On the other hand, IL-17 negatively correlated with neonatal birth weight. The concentrations of PLGF were significantly lower and sENG significantly higher in studied patients when compared to controls. The PlGF/sENG ratio in the PE group was significantly lower when compared to healthy third trimester pregnant patients. In the study group, negative correlations were observed between the sENG concentrations and thrombocyte levels. The higher concentrations of IL-17 in PE could suggest its role as an inflammatory agent in the pathogenesis of the syndrome. Moreover, the negative correlation between IL-17 and a neonatal birth weight could suggest the role of the cytokine in the development of fetal growth restriction (FGR) associated with PE. It seems possible that IL-17 can be a useful marker of the risk of FGR in pregnancy complicated by PE. Furthermore, the results suggested the potential role of sENG and the PlGF/sENG ratio in the prediction of adverse outcomes such as HELLP syndrome and DIC.

## 1. Introduction

Pre-eclampsia (PE) is a leading cause of perinatal morbidity and mortality associated with pregnant women and newborns in developed as well as developing countries [1,2]. Placental insufficiency associated with PE is dangerous for the fetus, leading to fetal growth restriction (FGR), the premature abruption of placenta and hypoxia, often causing stillbirth [3,4]. It has been observed that placental insufficiency in PE is associated with a vascular dysfunction. Potential abnormalities concerning vascular dysfunction include incomplete spiral artery remodeling, impaired placentation and, finally, endothelial damage. These abnormalities can be associated with immune alterations and the imbalance in pro- and antiangiogenic factors [5]. Moreover, some authors have underlined the suspicion of the immaturity of the pre-eclamptic placenta [6]. Currently, in the pathogenesis of PE, the role of inflammation is emphasized [7,8]. In the course of PE, an inappropriate trophoblast invasion occurs, which leads to the production of inflammatory mediators through the ischemic placenta and to the imbalance between pro- and antiangiogenic factors [7,8]. Thus far, no effective predictions of severe PE complications have been described that would allow for the prevention and better management of severe complications and adverse outcomes of the syndrome.

Recently, in order to better explore and understand the immunological complications of pregnancy, the Th1/Th2 paradigm was extended to the Th1/Th2/Th17/Treg paradigm [9]. Th17 cells have been discovered as a subpopulation of T cells, whose cytokine profile is different from Th1 and Th2 cells [10]. The main task of Th17 helpers is the production of interleukin-17 (IL-17). Many studies found an increased proportion of Th17 subpopulations in pregnancies complicated by miscarriage and preterm birth, as well as pre-eclampsia [11,12,13]. Interleukin-17 (IL-17, also known as IL-17A) is a major, strongly proinflammatory cytokine produced by Th17 helper cells [14]. Interleukin-17, a cytokine with potent pro-inflammatory properties, has a proven role in the development of inflammatory processes and acute immunological graft rejection, as well as autoimmune diseases. It has also been shown to affect the maturation of dendritic cells and inhibit the response from regulatory T cells (Treg), responsible for the phenomenon of immune tolerance [14].

The production of inflammatory mediators through the ischemic placenta and the release of soluble angiogenic and antiangiogenic factors into the maternal plasma seem to play a crucial role in the pathogenesis of PE. The proper trophoblast invasion and development of the placenta need an intensive angiogenesis. The process of angiogenesis, under normal physiological conditions, requires a dynamic balance between stimulants and inhibitory components [15,16,17,18].

The placental growth factor (PlGF) belongs to the family of vascular endothelial growth factors. PlGF plays a key role in the angiogenesis, proliferation, maturation and stabilization of new blood vessels. The receptor for PlGF is called fms-related tyrosine kinase-1 (VEGFR-1, sFlt-1). It is found on endothelial cells, smooth muscle cells and monocytes. The receptor sFlt-1 binds to PlGF and VEGF and reduces their serum concentrations. This way, the receptor blocks the interaction of VEGF with VEGFR-2 receptors, which are present on endothelial cells, and they are responsible for the synthesis of nitric oxide [19,20,21].

Endoglin (sENG) is an important antiangiogenic factor. Endoglin (CD105) is a type I membrane glycoprotein located on cell surfaces. It is a homodimer protein that consists of two subunits (95 kDa) joined with a disulfide bridge. It is localized on proliferating, active endothelial cells, and is present on syncytiotrophoblast cells [22,23]. It plays a crucial role in angiogenesis, in the development of the cardiovascular system and in vascular remodeling. Human endoglin is also involved in platelet–endothelium interactions [24]. Upon the proteolytic processing of the extracellular domain of ENG by matrix metalloproteinase-14 (MMP-14) and matrix metalloproteinase-12 (MMP-12), a circulating form of endoglin (soluble endoglin, sENG) can be released. Soluble endoglin can be detected in blood circulation [25,26].

The identification of the most useful inflammatory and angiogenic markers of pre-eclampsia is currently the subject of many studies. The purpose of our study was to estimate the role of IL-17, PlGF and sENG, as well as the PlGF/sENG ratio in pregnancy complicated by PE and in normal pregnancy, as well as their associations with severe complications and adverse outcomes of PE.

## 2. Materials and Methods

The patients participating in the study were admitted to the Department of Gynecology and Obstetrics, University Hospital, Rzeszow, Poland. The study group included 35 pregnant patients with pre-eclampsia (PE). The patients with PE were diagnosed according to the diagnostic criteria of the American College of Obstetricians and Gynecologists [27]. Blood samples from patients with PE were taken between the 29th and 35th week of pregnancy. None of patients from the study and control groups were affected by pre-existing clinical disorders. The exclusion criteria were chronic hypertension, renal diseases and autoimmune diseases, such as diabetes, lupus or rheumatoid arthritis, before pregnancy, as well as obesity. All pregnancies from the study and control groups were single. The patients with multiple gestations were excluded from the study and control groups because of difficulties with the interpretation of the results of the angiogenic factor concentrations in cases of multiple pregnancies when compared to single pregnancies. The patients from the study and control groups were not in active labor. The control group included 45 healthy women with uncomplicated pregnancies. They were recruited from the outpatient clinic. A flow chart shows the pregnant women from the control group that were included and excluded from the study (Figure 1).

Blood samples from healthy women were taken in the first (8–12 week), second (18–22nd week) and third trimester (28–34th week) of pregnancy. All patients gave their informed consent for inclusion before they participated in the study and for peripheral blood sampling. The study was conducted in accordance with the Declaration of Helsinki. The study design and protocol were approved by the local ethics committee (Figure 1). The clinical characteristics of patients from the study and control groups are presented in Table 1.

The sera concentrations of IL-17, PlGF and sENG were measured with the use of an immunoenzymatic method. Commercially available kits were used for the study. For the detection of IL-17, the Diaclone Company (Besancon, France) was used (catalog no. 850.940.096). For PlGF detection, an R&D Systems™ Human PlGF Quantikine ELISA Kit was used (catalog no. DPG00). For soluble endoglin detection, a Human CD105/soluble Endoglin ELISA Kit USCN Life Science Inc. (West Logan, UT, USA) (product no. SEA980Hu) was used. The measurements were performed according to the manufacturer’s protocols.

The results were presented as the mean and the standard deviation. The statistical differences between the groups were estimated using the Mann–Whitney U-test, chi-squared test and Fisher’s exact test. Differences were defined as statistically significant at the level of *p* < 0.05. For the correlation analysis, Spearman’s rank correlation coefficient test was performed. Two-tailed *p*-values of less than 0.05 were considered as statistically significant.

## 3. Results

The concentrations of IL-17 in sera of patients with pregnancies complicated by pre-eclampsia were significantly higher when compared to third trimester healthy pregnant women (PE vs. III trimester: IL-17: 3.90 ± 1.35 pg/mL vs. 2.35 ± 0.53 pg/mL; *p* < 0.01). In the third trimester of pregnancy, the concentrations were significantly higher when compared to the second trimester (III trimester vs. II trimester: 2.35 ± 0.53 pg/mL vs. 1.63 ± 0.56 pg/mL; *p* < 0.05). Furthermore, in the second trimester of pregnancy, the concentrations were significantly higher when compared to the first trimester (II trimester vs. I trimester: 1.63 ± 0.56 pg/mL vs. 1.12 ± 0.42 pg/mL; *p* < 0.05). The concentrations of IL-17 in the control group were higher the later a patient was included in the research and the later the blood was drawn for the analysis (R = −0.45, *p* < 0.05). This relationship suggested that in normal pregnancy, the concentration of IL-17 gradually increases. The results are presented in Figure 2.

In the group of patients with pre-eclampsia, the levels of IL-17 positively correlated with systolic blood pressure (R = 0.42; *p* < 0.01). On the other hand, the concentrations of IL-17 correlated negatively with the neonatal birth weight (R = −0.47, *p* < 0.01).

The concentrations of PlGF were significantly lower in patients with PE when compared to the controls (PE vs. III trimester: 3.36 ± 0.54 pg/mL vs. 29.24 ± 5.82 pg/mL, *p* < 0.0001). The concentrations of PlGF in the third trimester of a normal pregnancy were significantly higher when compared to the second trimester (III vs. II trimester: 29.24 ± 5.82 pg/mL vs. 21.65 ± 3.75 pg/mL, *p* < 0.05). Moreover, the concentrations of PlGF in the second trimester of normal pregnancy were significantly higher when compared to the first trimester (II vs. I trimester: 21.65 ± 3.75 pg/mL vs. 5.42 ± 0.89 pg/mL, *p* < 0.001). The results are presented in Figure 3.

In the group of patients with PE, moderate positive correlations were noticed between the concentrations of PlGF and the level of plasma protein (R = 0.47, *p* < 0.05). On the other hand, there were weak negative correlations between the concentrations of PlGF and diastolic pressure (R = −0.37, *p* < 0.01), as well as the concentrations of PlGF and proteinuria in the group of patients with PE (R = −0.65, *p* < 0.01).

In normal pregnancy, we observed positive correlations between the concentrations of PlGF and the week of pregnancy when the blood was taken (R = 0.8, *p* < 0.01).

The concentrations of sENG in patients with PE were significantly higher when compared to healthy women in the third trimester of normal pregnancy (PE vs. III trimester: 11.47 ± 4.65 ng/mL vs. 5.68 ± 2.78 ng/mL; *p* < 0.01). There were no statistically significant differences between the concentrations of sENG in the third and second as well as between the second and first trimesters of normal pregnancy (III vs. II trimester: 5.68 ± 2.78 ng/mL vs. 6.35 ± 2.52, ng/mL, *p* < 0.09; II vs. I trimester: 6.35 ± 2.52 ng/mL vs. 7.52 ± 3.58 ng/mL; *p* < 0.08). The results are presented in Figure 4.

In the PE group, there were weak positive correlations observed between the sENG concentrations and D-dimer levels (R = 0.35, *p* < 0.01). On the other hand, in the study group, there were moderate negative correlations between the sENG concentrations and thrombocyte levels (R = −0.45, *p* < 0.05).

In the study group, the PlGF/sENG ratio was significantly lower when compared to the PlGF/sENG ratio of healthy women in the third trimester of normal pregnancy (PE vs. III trimester: 0.34 ± 0.15 vs. 6.02 ± 2.75; *p* < 0.0001).

The ratio of PlGF/sENG in the third trimester was significantly higher when compared to the second trimester (III vs. II trimester: 6.02 ± 2.75 vs. 3.76 ± 2.04; *p* < 0.001). Similarly, there were statistically significant differences in the PlGF/ENG ratio between the second and first trimesters in normal pregnancy (II vs. I trimester: 3.76 ± 2.04 vs. 0.58 ± 0.22; *p* < 0.001). The results are presented in Figure 5.

## 4. Discussion

The mechanisms that maintain the balance of Th1/Th2 and Th17/Treg cells conditioning the normal development of the placenta and pregnancy are not fully understood.

It has been observed that in pregnancies complicated by PE, there is a deficit of Treg cells, which support the expression of Th17 lymphocytes and the induction of the inflammatory response in the feto–maternal interface [28,29].

In the current research, we observed increased concentrations of IL-17 in pregnancy complicated by PE. These observations were in line with other results, which showed a deficit in Treg cells and an increase in the population of Th17 cells, reported in pregnancy complications [10,29]. On the other hand, during normal pregnancy, the expansion of Treg cells with the decreased expression of Th17 cells was observed [10,29]. The higher concentrations of proinflammatory IL-17 in PE could suggest its role as a pathogenetic agent in the development of pre-eclampsia as an inflammatory state.

Furthermore, the results of our study showed that, in normal pregnancy, the concentrations of IL-17 gradually increased. Interestingly, Martinez-Garcia et al. also noted an increase in the level of IL-17 in the third trimester of uncomplicated pregnancy. The authors attributed an increase in the proinflammatory cytokine release near the term of delivery to the dilation of the cervix and the progress of labor [30].

Moreover, we observed that, in the PE group, the concentrations of IL-17 positively correlated with systolic blood pressure. The influence of IL-17 on blood pressure was explored in the study of Dhillion et al. The authors observed that the administration of IL-17 to healthy pregnant rats resulted in a statistically significant increase in mean arterial blood pressure. The administration of IL-17 to nonpregnant rats had no effect on blood pressure. Interestingly, the increase in blood pressure in pregnant rats was reversible after the administration of superoxide dismutase or the inhibition of angiotensin II receptor type 1, suggesting that the blood pressure was as a result of oxidative stress and the formation of autoantibodies against angiotensin II receptor type one [31].

Moreover, in PE, we observed a negative correlation between IL-17 and neonatal birth weight. This finding could indicate the role of IL-17 in the development of fetal growth restriction (FGR) associated with PE. Our observations suggested that IL-17 can be a useful marker for the risk of FGR, which is a severe fetal complication of PE.

We observed increased concentrations of sENG in pregnancy complicated by PE. There have been hypotheses that increased concentrations of sENG can play a role in the pathogenesis of pre-eclampsia, and are associated with maternal vascular dysfunction [32]. Robinson et al. observed that concentrations of sENG started to increase in the second trimester of pregnancy before the clinical symptoms of PE [33]. However, the authors suggested that the combination of pro- and antiangiogenic factors could characterize PE better than any single marker [34].

Panusunan-Lubis N et al. found that the assessment of diastolic notches in uterine arteries alone could not predict the incidence of early-onset PE. They concluded that PlGF levels and the pulsatile index (PI) of uterine arteries could be used as predictors of early-onset PE, although the examination of PlGF levels alone was sufficient as a predictor of early-onset PE [35]. These results suggested that Doppler velocimetry monitoring (PI and the assessment of the diastolic notch) could not be sufficient in the prediction and monitoring of PE. Furthermore, they emphasized the role of biochemical markers (especially PlGF) in the prediction and monitoring of this condition. Moreover, Chaiworapongsa et al. found that the incorporation of biochemical markers (sENG, PlGF and sVEGFR-1) significantly improved the risk assessment for these outcomes beyond that of clinical factors and the uterine and umbilical artery Doppler velocimetry [36]. Undoubtedly, the prediction and monitoring of PE with the use of Doppler velocimetry (PI and the presence of the diastolic notch in uterine arteries) have come to be a standard in maternal–fetal medicine. However, we should consider the fact that the diastolic notch was absent in the uterine arteries of 50% of pregnant women who later developed PE. On the other hand, approximately 50% of mothers with a bilateral diastolic notch did not later develop PE [37]. To take these facts into consideration, the role of biochemical markers in the prediction and monitoring of PE seems to be fundamental.

We observed negative correlations between the concentrations of sENG and thrombocyte levels in pregnancy complicated by PE. These results suggested the potential role of sENG in the prediction of increased risks of adverse PE outcomes, such as HELLP syndrome (hemolysis, elevated liver enzymes and low platelets) and disseminated intravascular coagulation (DIC).

The relationships of sENG with the severity of PE, clinical, and laboratory parameters, and the occurrence of adverse outcomes were not fully explained. Venkatesha et al. found that higher concentrations of sENG correlated with a severe course of PE [38]. In the study of Leanos-Miranda et al., the concentrations of sENG correlated positively with blood pressure, proteinuria and levels of creatinine, uric acid, aspartate aminotransferase, alanine aminotransferase and lactate dehydrogenase, as well as, inversely, with gestational age, infant birth weight and platelet counts [39].

Positive correlations between the coagulation parameters and the severity of PE were also observed in the study of Van Walraven C et al. The authors in their study observed an increased risk of subsequent thromboembolism in the group of patients with PE [40]. Similar results were obtained by Oladosu-Olayiwola et al. [41]. They found abnormal levels of D-dimer, plasminogen activator inhibitor-1 (PAI-1) and tissue plasminogen activator (tPA) in patients with PE, suggesting that monitoring the levels of these parameters could help in the management of the condition [41]. Hammerova et al. observed a significantly shorter prothrombin time, international normalized ratio (INR) and activated partial thromboplastin time (aPTT), significantly higher plasma concentrations of D-dimer and fibrinogen and higher activity of factor VII and X in women in third trimester pregnant, with adverse pregnancy outcomes [42]. On the other hand, prospective studies revealed increased t-PA levels and decreased PAI-2 levels in pregnant women who later developed PE [43,44].

We are aware of some limitations of our study. The limitations of the study included the relatively small number of patients included in the studied group. On the other hand, the strengths of our study included the lengthy clinical characteristics of the groups, as well as detailed laboratory result descriptions. The real advantages were associations found between serious complications of pre-eclampsia and patient’s laboratory as well as clinical findings.

## 5. Conclusions

The higher concentrations of IL-17 in PE might suggest its role as an inflammatory agent in the pathogenesis of disease. Moreover, the negative correlation between IL-17 and neonatal birth weight could suggest the role of IL-17 in the development of fetal growth restriction (FGR) associated with PE. In conclusion, IL-17 could be a useful marker for determining the risk of FGR in pregnancy complicated by PE. Furthermore, the sENG concentrations and PlGF/sENG ratio could suggest their potential roles in the prediction of increased risks of adverse outcomes, such as HELLP syndrome and DIC.

## Figures and Tables

**Figure 1 ijerph-20-00768-f001:**
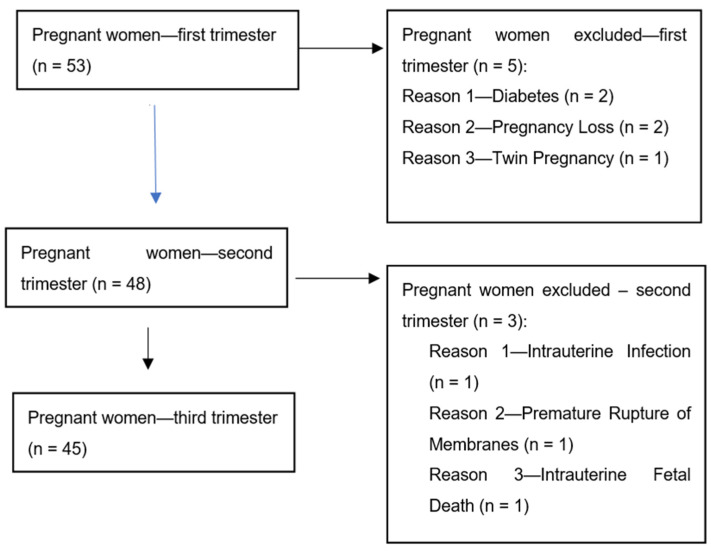
Flow chart showing the pregnant women from the control group that were included and excluded from the study.

**Figure 2 ijerph-20-00768-f002:**
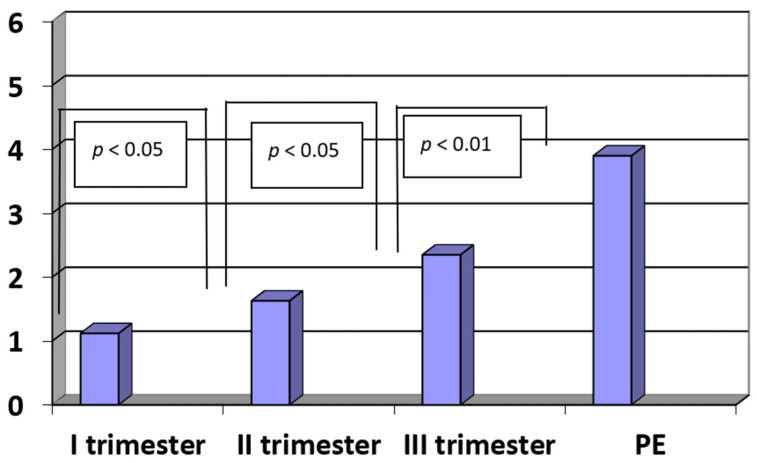
The concentrations of interleukin-17 (IL-17) (pg/mL) in sera of patients with PE (n = 35) and in sera of women in the first, second and third trimesters of normal pregnancy (n = 45).

**Figure 3 ijerph-20-00768-f003:**
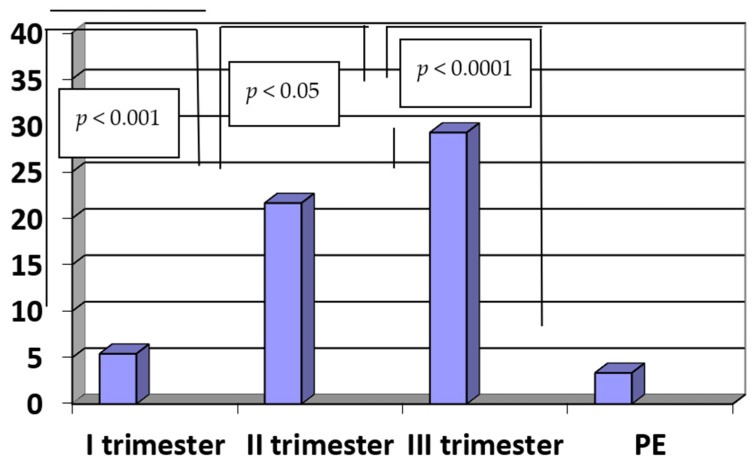
The concentrations of PlGF (pg/mL) in sera of patients with PE (n = 35) and in sera of women in the first, second and third trimesters of normal pregnancy (n = 45).

**Figure 4 ijerph-20-00768-f004:**
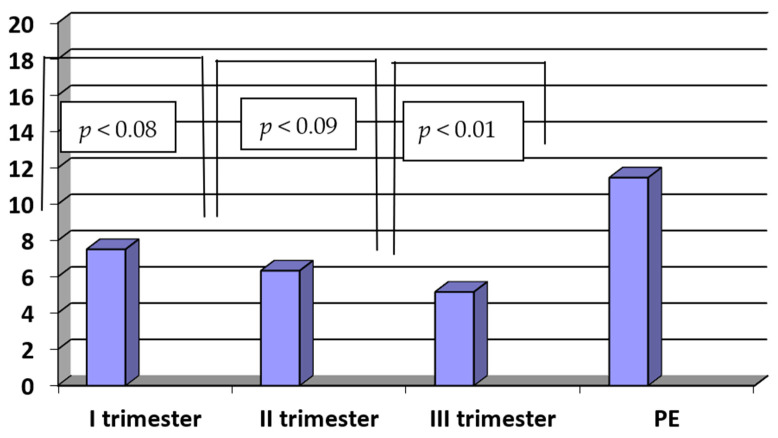
The concentrations of sENG (ng/mL) in sera of patients with PE (n = 35) and in sera of women in the first, second and third trimesters of normal pregnancy (n = 45).

**Figure 5 ijerph-20-00768-f005:**
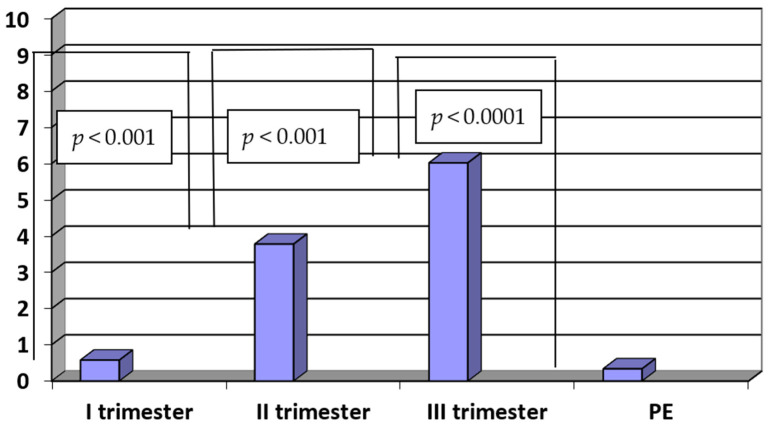
The ratio of PlGF/sENG in sera of patients with PE (n = 35) and in sera of women in the first, second and third trimesters of normal pregnancy (n = 45).

**Table 1 ijerph-20-00768-t001:** The clinical and laboratory characteristics of pregnant patients with PE (n = 35) and women in normal pregnancy (n = 45).

	I Trimester	II Trimester	III Trimester	PE	Statistical Significance
Age (years)	30 ± 5	30 ± 5	30 ± 5	33 ± 4	NS
First pregnancy	17	17	17	21	NS
Subsequent pregnancy	27	27	27	12	NS
The duration of gestation (days)	-	-	275 ± 18	245 ± 35	<0.005
Time of blood collection (weeks of gestation)	10 ± 2	20 ± 2	32 ± 2	32 ± 3	NS
Birth weight (g)	-	-	3210 ± 375	2278 ± 976	<0.0001
Systolic blood pressure (mmHg)	120 ± 25	110 ± 15	112 ± 13	155 ± 14	<0.001
Diastolic blood pressure (mmHg)	85 ± 10	78 ± 15	73 ± 9	102 ± 10	<0.001
Total protein (g/dL)	7.6 ± 0.5	7.5 ± 0.4	6.4 ± 0.3	5.8 ± 0.65	<0.01
Pulsatile index (PI) in umbilical artery (UA)	-	-	0.85 ± 0.25	0.62 ± 0.20	<0.05
Proteinuria (g/L)	-	-	-	545 ± 130	-
ALAT (U/L)	18 ± 6	21 ± 5	22 ± 8	44 ± 12	<0.05
Platelets (10^3^/μL)	250 ± 65	240 ± 55	233 ± 60	205 ± 75	<0.05
Prothrombin time (s)	12.0 ± 0.4	11.0 ± 0.4	11.0 ± 0.4	10.2 ± 0.5	<0.05
D-dimer (μg/L)	457 ± 180	565 ± 210	957 ± 240	1567 ± 1100	<0.05
Fibrinogen (g/L)	3.8 ± 0.8	4.5 ± 0.7	4.9 ± 0.9	4.5 ± 1.1	NS
Kalium (mmol/L)	3.8 ± 0.4	3.9 ± 0.3	4.0 ± 0.3	4.2 ± 0.4	NS
Natrium (mmol/L)	137 ± 1.5	135 ± 1.4	135 ± 1.6	138 ± 3.4	NS
Glomerular filtration rate (mL/min/1.73 m^2^)	101 ± 30.5	102 ± 28.5	104 ± 32.3	87.8 ± 37.8	<0.05
Creatinine (mg/dL)	0.5 ± 0.3	0.5 ± 0.4	0.6 ± 0.3	0.8 ± 0.3	NS
Uric acid (mg/dL)	3.8 ± 0.5	3.9 ± 0.5	3.9 ± 0.8	5.9 ± 1.6	<0.005
Urea (mg/dL)	16.1 ± 4.8	17.8 ± 5.0	19.1 ± 5.2	22.45 ± 15	NS

## Data Availability

The data presented in this study are available on request from the corresponding author.

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
