# Peer review of "The Association of IL-17 and PlGF/sENG Ratio in Pre-Eclampsia and Adverse Pregnancy Outcomes"

_ijerph, 2022, doi:10.3390/ijerph20010768_

Round 1

Reviewer 1 Report

I am pleased to have the possibility to review the study “The association of IL-17 and PlGF/sENG ratio in preeclampsia and adverse pregnancy outcomes”. The discussed problem of preeclampsia has a meaningful impact on maternal and children’s health. The possible prediction of preeclampsia occurrence or its severity assessment is nowadays the topic of many discussions as the need to improve fetal well-being and neonatal outcomes because of the potential possibility of using Aspirin in Preeclampsia prophylaxis. The study population and the character of the study are undoubtful study strengths. Nevertheless, several minor issues require clarification before publication.

The introduction section is a well-constructed part of the study. The problem was appropriately introduced to the reader, and the aim of the study was clearly presented. I also have no remarks about the Methodological part and used statistical methods. The study was conducted very close to PRISMA guidelines for observational studies. The PICO question is well described, and the inclusion and exclusion criteria are well presented. Nevertheless, I strongly recommend adding the PRISMA flow chart as a supplementary file.

The result section is described in an appropriate way. Nevertheless, I suggest replacing statistical data named “NS” with numerous p-values.

The discussion section is performed appropriately. The study's objectives were answered, and a comparison to the literature was made. In this section, only a description of the limitations and strengths of the study is missing. I recommend adding it as the last paragraph before the conclusions.

Author Response

Dear Sir/Madam,

Thank you very much for your constructive comments. 

I have made the changes according the suggestions of the Reviewer 1. 

  1. In the "Introduction" section - PRISMA flow chart has been added as a Figure 1.
  2. In the "Result" section statistical data named NS have been replaced with numerous p-values. 
  3. In the "Discussion" section - the description about limitations and streghts of the study have been added in the last paragraph before the Conclusions. 

Thank you once again for the constructive comments. 

Sincerely yours, 

Dorota Darmochwal-Kolarz

Reviewer 2 Report

this is a very interesting manuscript assessing the role of concentrations of Interleukin-17 (IL-17), Placental Growth Factor (PlGF), soluble Endoglin (sENG), as well as PlGF/sENG ratio in pregnancy complicated by pre-eclampsia (PE) and normal pregnancy.

Although the manuscript is interesting and generally well written, some revisions are needed. In particular:

lines 29-36: although authors properly introduced the clinical manifestation of preeclampsia, it deserves to be pointed out that preeclampsia is also charactherized by  trophoblast immaturity (PMID: 32529396)  and vascular dysfunction (PMID: 34831277). This is an important point to highlight since they could also be the cause of the alterations found by the authors further highlithing their interesting results.

2. Materials and Methods: the product codes of ELISA kits used must be reported

Table 1: authors must report how many samples collected for each group (first, second and third trimester) and their demographic and clinical characteristics (maternal age, gestational week...). Moreover, the gestational age in PE pregnancies is lower compared to normal pregnancies, since IL-17 and PlGF levels are related to the gestational age authors must compare only age-matched pregnancies otherwise the altered values found could be due to the lower gestational age rather than preeclampsia. This also explain the lower birth weights found in PE group. Did the authors include PE complicated by FGR?

Author Response

Dear Sir/Madam,

Thank you very much for the constructive comments. 

Below there are answers for the Reviewer 2. 

  1. In the "Introduction section" (lines 29-36) it has been pointed now that preeclampsia is characterized by vascular dysfunction and trophoblast immaturity.
  2. In the "Materials and Methods section" the product code of ELISA kit have been now reported. 
  3. Samples, demographic and clinical characteristics of pregnant women have been now reported. 
  4. In our study preeclamptic patients have been compared to gestational age-matched pregnancies.
  5. In the group of patients witrh PE fetal weight were calculated between 10-95 percentile. 

Thank you very much once again for the comments. 

Sincerely yours, 

Dorota Darmochwal-Kolarz

Round 2

Reviewer 2 Report

the manuscript has been significantly improved and can be accepted in the present form